# A Multifunctional and Fast-Response Lysosome-Targetable Fluorescent Probe for Monitoring pH and Isoxaflutole

**DOI:** 10.3390/ijms23116256

**Published:** 2022-06-02

**Authors:** Liu Yang, Yan Liu, Mingli Yue, Ping Li, Yulong Liu, Fei Ye, Ying Fu

**Affiliations:** Department of Applied Chemistry, College of Arts and Sciences, Northeast Agricultural University, Harbin 150030, China; yangliu@neau.edu.cn (L.Y.); liuyansjm@163.com (Y.L.); yuemingli@neau.edu.cn (M.Y.); liping@neau.edu.cn (P.L.); liuyulong@neau.edu.cn (Y.L.)

**Keywords:** fluorescence, pH sensing, lysosome target probe, isoxaflutole, ICT

## Abstract

A new chemosensor, namely *N*-(2-morpholinoethyl)acetamide-4-morpholine-1,8-naphthimide (**MMN**), was designed and synthesized through an amidation reaction. **MMN** was fabricated as a multifunctional fluorescent probe for monitoring pH and isoxaflutole. **MMN** exhibited excellent stability in MeCN/H_2_O (*v*/*v*, 9/1), with an obvious “off–on” fluorescence response toward pH changes due to intramolecular charge transfer (ICT), where the linear response ranges of **MMN** in the weakly acidic system were from 4.2 to 5.0 and from 5.0 to 6.0 with apparent p*K_a_* = 4.62 ± 0.02 and 5.43 ± 0.02. Based on morpholine as the lysosome targetable unit, **MMN** could selectively locate lysosomes in live cells. **MMN** also successfully detected the presence of H^+^ in test papers. Finally, **MMN** could specifically recognize isoxaflutole at a detection limit of 0.88 μM. A possible sensing mechanism was identified based on density function theory calculations. These results indicate that **MMN** could be a superior potential chemosensor for detecting pH and isoxaflutole selectively and sensitively and could be used in real sample detection.

## 1. Introduction

pH is essential to human life and plays vital roles in some physiological and chemical processes, including enzymatic activity, cell function, environmental monitoring and human tissue fluid [1]. Intracellular pH homeostasis is necessary for the regulation of various basic cellular processes to maintain normal cell function [2,3,4]. Lysosomes, as essential acidic subcellular organelles (pH 4.5–5.5) in eukaryotic cells, participate in intracellular digestion and cellular differentiation [5,6]. Monitoring lysosome pH change is very important in the development of medical diagnoses and treatment methods. Hence, exploring a more effective strategy for the accurate detection of pH is crucial.

In the past few years, various techniques have been used for pH detection such as the colorimetric method [7], electrochemistry procedures [8], ion-sensitive field-effect transistors [9] and fluorescence techniques [10]. However, there are still shortcomings such as photobleaching, high background emission, alkalization, complicated sample preparation, the need for sophisticated instruments, and high time-consumption and costs. Despite these limitations, fluorescent techniques have the advantages of being convenient, efficient, sensitive and selective and have the potential to be used for monitoring pH.

Pesticides refer to chemical agents used in agriculture to control pests and to regulate plant growth. Pesticides are widely used in agriculture, forestry and animal husbandry; in environmental and family hygiene, pest control and epidemic prevention; and in industrial products to prevent mildew and moths [11]. Isoxaflutole is a highly effective preseedling herbicide that is used in maize, sugarcane and other dry crop fields [12,13,14]. However, the long-term application of pesticides across wide areas not only bring potential health risks to humans but also cause environmental pollution [15,16,17]. The negative impact of pesticide residue has thus aroused people’s attention.

Current reports on the detection methods for isoxaflutole residue mainly involve liquid chromatography and liquid chromatography-tandem mass spectrometry [18,19]. Even though the chromatography method has the advantages of being highly sensitive, having a superior separation ability and having excellent selectivity, this method has a long pretreatment time and can have interferences in the detection process. It can also be costly and requires the presence of highly selective detectors for detecting pesticides. A more accurate, fast, efficient and convenient method in the detection of pesticides is needed. Chemosensors such as fluorescent probes have the potential to detect isoxaflutole with the advantages of low costs, easy operation, fast signal response and signal visualization.

A new lysosome-targetable pH fluorescent probe, namely *N*-(2-morpholinoethyl)acetamide-4-morpholine-1,8-naphthimide (**MMN**), was synthesized and characterized. **MMN** was synthesized by introducing the morpholine unit [20,21,22] as a lysosomal targeting group to naphthalimide, which was highly sensitive to pH changes due to intramolecular charge transfer (ICT) [23]. The fluorescence intensity of **MMN** increased linearly to 2.2 times in the apparent pH range from 4.2 to 6.0, and the apparent p*K_a_* was 4.62 ± 0.02 and 5.43 ± 0.02. The proposed sensing mechanism was confirmed by ^1^H NMR. **MMN** selectively located lysosomes in live cells and successfully detected H^+^ in test papers. Additionally, **MMN** could specifically recognize isoxaflutole at a detection limit of 0.88 μM.

## 2. Results and Discussion

### 2.1. Design and Synthesis of **MMN**

Naphthalimide, as a representative of fluorophore, shows a high fluorescence quantum yield and high thermal stability [24,25]. Morpholine has been recognized for its special lysosomal targeting function [26]. Based on a rational design, **MMN** was synthesized through an amidation reaction, as illustrated in Figure 1. The fluorescent probe was characterized by FTIR, ^1^H NMR, ^13^C NMR and HRMS spectroscopy. **MMN** exhibited an obvious “off–on” fluorescence response toward pH changes and detected isoxaflutole.

### 2.2. pH-Dependent Sensing Performance

The solvent effect of **MMN** was studied through fluorescence measurements in different solvents (Appendix A). Based on the solvent fluorescence response and a low toxicity, the spectral properties of **MMN** were investigated in CH_3_CN/H_2_O (*v*/*v*, 9/1) where the apparent pH was 6.0. The fluorescence spectra of **MMN** exhibited a striking dependence on pH (Figure 1a). **MMN** showed a weak fluorescence signal at 529 nm in the strongly acidic apparent pH region (pH < 4.2). When the alkalinity of the solution increase to weakly alkaline (pH > 6.0), the fluorescence intensity increased significantly, with a slow blueshift by 6 nm. 

To further evaluate the dependence of **MMN** on pH, the UV–Vis spectra of **MMN** at different pH values were investigated. As the pH values of the solution increased, a blueshift of the UV absorption band was observed (Figure 1b). **MMN** displayed a superior “off–on” switching behavior from apparent pH 2 to 10 and demonstrated an obvious change in the electron density of naphthalene rings, which was consistent with the fluorescent emission spectra analyses. The fluorescence spectra of **MMN** were investigated to examine its anti-interference in the presence of other ions at a representative pH value (pH = 4.2). As shown in Appendix A, **MMN** possesses the potential to accurately detect pH changes with negligible interference.

### 2.3. pK_a_ Value, Reversibility and Photostability

As shown in Figure 2a, the fluorescence intensity of **MMN** expressed an excellent partitioned response to pH, i.e., the partitioned linear ranges of apparent pH were from 4.2 to 5.0 with a correlation coefficient (*R*^2^) of 0.99 and from 5.0 to 6.0 with an *R*^2^ of 0.99. The fluorescence intensity of **MMN** with different apparent pH values was used to calculate the acidity-constant apparent p*K_a_* values of **MMN** based on the Henderson–Hasselbalch equation in MeCN/H_2_O (*v*/*v*, 9/1):$$pK_{a}=pH-log\left(\frac{I_{max}-I}{I-I_{min}}\right)$$
where *I* is the observed fluorescence intensity at a fixed wavelength, and *I_max_* and *I_min_* are the corresponding maximum and minimum intensities, respectively [27]. These results indicated that **MMN** exhibited a high sensitivity to weakly acidic pH with apparent p*K_a_* values of 4.62 ± 0.02 and 5.43 ± 0.02. The reversibility behavior of the prepared **MMN** probe was investigated by tracking the changes in emission intensity at 529 nm at apparent pH 4.2 and pH 6.0 (Figure 2b). The changes in colors (inset of Figure 2b) were recorded at apparent pH 4.2 and 6. There was no significant change in the reversibility up to four cycles. Thus, this proved that **MMN** could be used as a reversible pH monitor due to its excellent reversibility. In Figure 2c, the response of the proposed **MMN** to H^+^ was in real time, and the fluorescence intensity remained nearly unchanged over 130 min when tested at apparent pH 4.2 and 6.0. The results indicated that **MMN** could be reliably applied for real-time monitoring of pH in practical applications.

### 2.4. Recognition Mechanism of MMN to H^+^

To examine the proposed interactive mechanism of **MMN** on the pH value, ^1^H NMR spectra of **MMN** (D_2_O/DMSO-*d_6_* (*v*/*v*, 1:1)) were measured by adding TFA (H^+^) or NaOH (OH^−^) to solutions. As shown in Figure 3a, upon the addition of H^+^ to **MMN**, the signal at δ 8.10–8.13 ppm was downfield-shifted to 8.45–8.47 ppm owing to a decrease in the electron density around N-H of amide induced by H^+^. The signal of the aromatic protons (Ar-H) was not affected except for a later peak fraction. However, the ^1^H NMR spectrum of **MMN + OH^−^** was consistent with that of **MMN**. Therefore, when H^+^ was added, the change in ^1^H NMR spectrum contributed to a decrease in electron density around N-H of amide and the active hydrogen protons NH produced a charge transfer process, where N atom was positively charged and C=N was formed due to tautomerism. When OH^−^ was added, the intramolecular charge transfer (ICT) effect was weakened, and a blueshift and enhancement of the fluorescent emission band were observed [23]. The mechanism of **MMN** for H^+^ is proposed in Figure 3b.

### 2.5. Colocalization Imaging Experiment and Test Stri

The colocalization experiment was carried out using a commercial lysosome-specific dye, Lyso-Tracker Red, to confirm the lysosome-targeting ability of **MMN**. As shown in Figure 4, **MMN** displayed blue punctuated fluorescence (Figure 4b), which merged well with the red fluorescence produced by Lyso-Tracker Red (Figure 4d) and had a high Pearson’s colocalization coefficient of 0.67 with an overlap coefficient of 0.99 (Figure 4e). Simultaneously, the intensity profile within the ROI in the blue and red channels displayed a trend of synchronization (Figure 4f). The results demonstrate that **MMN** could selectively locate lysosomes in live cells and has the potential to detect pH in lysosomal cells. This is evident in the test paper experiments shown in Figure 5. When the apparent pH decreased from 6.0 to 4.2, the fluorescent colors of the test papers changed from light green to ivory by degrees. The dye strip results show that **MMN** can be used for rapid and portable pH monitoring. In addition, the performance of **MMN** was compared with other previously reported pH probes (Table 1) [28,29,30].

### 2.6. Detection of Isoxaflutole

To further investigate the detection properties of **MMN, MMN** was placed in a CH_3_CN solution along with different pesticides, including triketones, fluorine, cyanide and isoxazole. In the fluorescence spectra (Figure 6a), the addition of isoxaflutole to the solution of **MMN** induced obvious fluorescence quenching, while the other pesticides exerted a negligible influence under the same conditions. Similarly, the **MMN** solution could be clearly observed only when isoxaflutole was added and the fluorescence changed from strongly green to quenched (inset in Figure 6a). When isoxaflutole was added to the probe solution, the absorption spectrum was not affected at 400 nm and a new strong absorption band appeared, centered at 293 nm. (Appendix A). The results indicate that **MMN** could be used to distinguish isoxaflutole from these other pesticides. To further illustrate the specific detection of isoxaflutole by **MMN**, the fluorescence spectra of **MMN** were investigated when other pesticides existed in the background. As shown in Figure 6b, no noticeable fluorescence interference from other pesticides was observed, revealing that **MMN** possesses high selectivity as a disturbance-free isoxaflutole fluorescent probe.

For a sensitivity analysis, a fluorescence titration study of **MMN** was conducted in the presence of various concentrations of isoxaflutole. The intensity of fluorescence at 529 nm decreased gradually with the addition of an increasing amount of isoxaflutole (Figure 7a). The emission intensity stabilized after the amount of added isoxaflutole ion reached 5 equiv. and showed a linear relationship (*R*^2^ = 0.98 or 0.99) with the concentration of isoxaflutole in the ranges of 10–27 or 28–50 µM (inset in Figure 7a). Moreover, based on the equation LOD = 3*σ/k*, the LOD of **MMN** for isoxaflutole was determined to be 0.88 µM, where *σ* is the standard deviation of the response at the lowest concentrations and *k* represents the slope of the calibration. The results indicate that **MMN** could be used to quantitatively determine isoxaflutole with a low detection limit. The binding constant of **MMN** with isoxaflutole was calculated according to the fluorescence intensity data using the modified Stern–Volmer equation: $I_{0}/\left(I_{0}-I\right)=1/A+1/K\cdot 1/\left[Q\right]$ [31], where *I*_0_ and *I* are the maximum luminescent intensities of **MMN** before and after adding isoxaflutole, *K* is the binding constant (M^−1^) and the unit measurement [A] represents molar concentration. The binding constant (*K*) was calculated as 3.7 × 10^5^ M^−1^ (Figure 7b), which was compared with other previously reported fluorescent probes (Appendix A). Considering that response time is a crucial factor for photostability, the time-dependent curve was studied, as shown in Appendix A. The results demonstrate that the fluorescence signal of free **MMN** remained stable while **MMN** showed immediate and distinct fluorescence quenching; the fluorescence intensity reached the minimum within 30 s and remained stable in the following 90 min, indicating a high reactivity of **MMN** with isoxaflutole.

A Job’s plot analysis was carried out to determine the stoichiometric ratios of **MMN** to isoxaflutole,. The emission intensity reached the maximum when the molar fraction of isoxaflutole was 0.5, which indicates that **MMN** and isoxaflutole act in a stoichiometric ratio of 1:1 (Appendix A). The possible mechanism of **MMN** for detecting pesticides was investigated by density functional theory (DFT) calculations exhibited via the electron density or energy level between isoxaflutole and **MMN**. As shown in Figure 8, several representative pesticide molecules for DFT were selected as examples for investigation. It was found that the LUMO energy levels of isoxaflutole and mesotrione were lower than those of other pesticides (glyphosate, oxyfluorfen, pyrazoxyfen and cypermethrin), indicating that the electron affinities of isoxaflutole with the -CF_3_ electron-withdrawing group and mesotrione with the -NO_2_ electron-withdrawing group were relatively higher than those of other pesticides. Meanwhile, their LUMO energy levels were between the HOMO and LUMO energy levels of **MMN**, which could cause the transition of electrons from the LUMO orbital of **MMN** to the LUMO orbitals of isoxaflutole and mesotrione in the excited state [32,33,34,35,36,37]. The transition of electrons from the LUMO orbital of **MMN** in the excited state to the LUMO of isoxaflutole prevented the electrons from returning to the HOMO orbital of **MMN**, leading to the phenomenon of fluorescence quenching in the recognition of isoxaflutole. The electron-withdrawing group in isoxaflutole may play a vital role in quenching effects due to the photoinduced electron transfer from **MMN** to electron-deficient isoxaflutole. Therefore, DFT illuminates the possible mechanism of how **MMN** detects isoxaflutole.

## 3. Experimental Section

### 3.1. Materials and Methods

All reagents used in the experiment, including raw materials, solvents and pesticides, were purchased from commercial suppliers and used without purification. A Bruker ALPHA-T spectrometer (KBr, Bruker, Ettlingen, Germany) was used to record FTIR spectra. The ^1^H NMR and ^13^C NMR spectra of the samples were obtained using a Bruker AVANCE 400 MHz system (Bruker, Germany). High-resolution mass spectrometry (HRMS) was performed on an FTMS Ultra Apex MS spectrometer (Bruker Daltonics Inc., Billerica, MA, USA). UV–Vis and fluorescence spectra were measured on a UV-2550 ultraviolet spectrophotometer (Shimadzu, Kyoto, Japan) and a PerkinElmer LS55 fluorescence spectrometer (PerkinElmer, Buckinghamshire, UK), respectively. All pH values were measured with a PHS-3C pH meter (Inesa, Shanghai, China). Cell images were obtained on a LEICATCSSP2 confocal laser scanning microscope (Leica, Wetzlar, Germany).

### 3.2. Synthesis

#### 3.2.1. Synthesis of 4-Morpholine-1,8-naphthalic Anhydride (**1**)

Compound **1** was synthesized based on the published literature [38]. 4-Bromo-1,8-naphthalic anhydride (5.54 g, 20.0 mmol) was added to 2-methoxyethanol (25 mL) in a three-necked flask and stirred at 25 °C until it dissolved. Then, 1.92 mL (22.0 mmol) of morpholine was dropped into the reaction system and the temperature was increased to 125 °C, after which the system was refluxed for 5 h. After cooling to room temperature, the insoluble orange needle-like precipitate was separated out. The residue was recrystallized from EtOH to obtain the final yellow needles (1). Yield: 90%. m.p.: 227.7–228.5 °C. All spectra of the structural characterization of compound **1** are presented in the electronic Appendix A. FT-IR (KBr) cm^−1^: 3076, 2954, 2852 (*v* C-H), 1762, 1724 (*v* C=O). ^1^H NMR (CDCl_3_, TMS, 400 MHz, ppm) δ 8.61 (m, 1H), 8.55 (d, *J* = 8.1 Hz, 1H), 8.48 (m, 1H), 7.76 (m, 1H), 7.28 (s, 1H), 4.07–4.00 (m, 4H), 3.35–3.29 (m, 4H). ^13^C NMR (CDCl_3_, TMS, 100 MHz, ppm): 156.83, 134.88, 133.32, 131.57, 126.18, 115.26, 77.22, 66.83, 53.31.

#### 3.2.2. Synthesis of *N*-Carboxymethyl-4-morpholine-1,8-naphthalimide (**2**)

Compound **2** was synthesized by improving the previous synthesis method [39]. A mixture of compound **1** (2.85 g, 10 mmol) and glycine (1.15 g, 15 mmol) was refluxed with continuous stirring in *N,N*-dimethylformamide (DMF) (75 mL) at 100 °C for 30 h. The filtrate was transferred to a water-filled beaker. After the yellow solid was completely separated out, the crude product was obtained by filtration, which was purified by recrystallization from ethanol to give compound **2**. Yield: 86%. m.p.: 247.7–248.4 °C. All spectra of the structural characterization of compound **2** are presented in the electronic Appendix A. FT-IR (KBr) cm^−1^: 3435 (*v* OH), 2960, 2821 (*v* C-H), 1735, 1701,1658 (*v* C=O). ^1^H NMR (DMSO-*d*_6_, TMS, 400 MHz, ppm) δ 13.04 (s, 1H), 8.57–8.42 (m, 3H), 7.85 (m, 1H), 7.39 (d, *J* = 8.1 Hz, 1H), 4.72 (s, 2H), 3.97–3.84 (m, 4H), 3.29–3.19 (m, 4H). ^13^C NMR (DMSO-*d*_6_, TMS, 100 MHz, ppm): 169.88, 163.75, 163.16, 156.35, 133.05, 131.58, 131.51, 129.71, 126.70, 125.79, 122.57, 115.67, 66.64, 53.51, 41.50.

#### 3.2.3. Synthesis of *N*-(2-Morpholinoethyl)acetamide-4-morpholine-1,8-naphthimide (**MMN**)

The compound *N-*(2-morpholinoethyl)acetamide-4-morpholine-1,8-naphthimide was prepared by adopting the reported procedure [40,41,42]. Compound **2** (5 mmol) was reacted with 4-(2-aminoethyl)morpholine by the coupling reagents 1-ethyl-3-(3-dimethylaminopropyl)carbodiimide (EDC, 1.1 eq), the base *N,N*-diisopropylethylamine (DIEA, 2 eq) and hydroxybenzotrizole (HOBt, 1.2 eq) in dry DMF. After 10 h, the reaction was quenched by adding water and the desired product was precipitated from the reaction mixture, which was filtered and dried. The mixture was purified by column chromatography on silica gel eluted with CH_2_Cl_2_/MeOH (*v*/*v*, 15/1). Yield: 75%. m.p. > 280 °C. All spectra of the structural characterization of compound **MMN** are presented in the electronic Appendix A. FT-IR (KBr) cm^−1^: 3305 (*v* NH), 3097, 2924, 2854 (*v* C-H), 197, 1662 (*v* C=O). ^1^H NMR (DMSO-*d*_6_, TMS, 400 MHz, ppm) δ 8.54 (d, *J* = 8.5 Hz, 1H), 8.50 (d, *J* = 7.3 Hz, 1H), 8.43 (d, *J* = 8.1 Hz, 1H), 8.12 (s, 1H), 7.85 (s, 1H), 7.39 (m, 1H), 4.63 (s, 2H), 3.93 (t, *J* = 4.6 Hz, 4H), 3.56 (t, *J* = 4.7 Hz, 4H), 3.29–3.14 (m, 6H), 2.35 (m, 6H). ^13^C NMR (DMSO-*d*_6_, TMS, 100 MHz, ppm): 167.14, 163.92, 163.37, 156.08, 132.73, 131.21, 129.77, 126.61, 125.78, 123.03, 116.26, 115.56, 66.65, 55.38, 53.70, 53.52, 42.74; HRMS (ESI): calcd. for C_24_H_28_N_4_O_5_ ([M+H] ^+^) 453.2060, found 453.2144.

### 3.3. General Method for the Spectra Experiments of **MMN**

The stock solution of **MMN** was prepared in CH_3_CN/H_2_O (*v*/*v*, 9/1). The 1.00 × 10^−2^ M stock solutions of metal cations (KCl, NaCl, AgNO_3_, PbCl_2_, SnCl_2_, CoCl_2_, CuCl_2_, HgCl_2_, BaCl_2_, MnCl_2_, CaCl_2_, ZnCl_2_, MgCl_2_, NiCl_2_, CrCl_3_, FeCl_3_ and AlCl_3_) were prepared with deionized water. The 1.00 × 10^−2^ M solutions of anions (F^−^, Cl^−^, Br^−^, I^−^, NO_3_^−^, NO_2_^−^, CN^−^, SCN^−^ H_2_PO_4_^−^, SO_4_^2−^ and HSO_4_^−^) were prepared by tetrabutylammonium (TBA) salts and sodium salts with deionized water. The ionic salts were dissolved in distilled water to prepare the stock solutions for the cations and anions (1.00 × 10^−2^ M). The stock pesticide solutions (1.00 × 10^−2^ M) were prepared in DMSO due to the poor solubility of some pesticides using cypermethrin, cyhalothrin, oxyfluorfen, mesotrione, cyhalofop-butyl, teflubenzuron, NTBC, sulcotrione, tembotrion, isoxaflutole, pyrazoxyfen, glyphosate, clethodim, fluazuron, flusilazole, fluorobenzene and benoxacor. For different pH solutions, 1.00 × 10^−1^ M HCl or NaOH was added to 10 mL of 1.00 × 10^−5^ M **MMN** solution and the apparent pH value was determined with a pH meter. To investigate the anti-interference of **MMN**, 50 μL aliquots of 1.00 × 10^−2^ M different cation and anion solutions were used as the background to measure the fluorescence spectra at two representative apparent pH values (4.2 and 6.0). Different instruments were used for fluorescence spectroscopic detection of pH and isoxaflutole. The slit widths of excitation and emission were set to 10 nm, and the excitation wavelength was set to 415 nm in the fluorescence spectral experiment.

### 3.4. Cell Incubation and Fluorescence Imaging

Human stromal cell line (HSC) cells were purchased from the Chinese Academy of Sciences and cultured using the medium DMEM, which contains 15% fetal bovine serum, 100 µg/mL penicillin and 100 g/mL streptomycin at 37 °C within a humidified 5% CO_2_ atmosphere. One day before the fluorescence imaging experiment, the cells were placed on petri dishes (NESTC) at the bottom of 35 mm cover slides. The cells were incubated with Lyso-Tracker Red (70 nM) for 30 min, washed with PBS 3 times, fixed with 4% paraformaldehyde for 1 h, washed with PBS 3 times, washed with a solvent 3 times, stained overnight with **MMN** (100 nM), washed with a solvent 3 times, sealed and observed under a confocal microscope.

### 3.5. Computational Methods

The Perdew–Burke–Ernzerh (PBE) method of Generalized Gradient Approximation (GGA) in the Dmol3 module was used to calculate the exchange correlation energy using a DNP+ basis set calculation and dispersion correction with Grimme [43,44,45].

## 4. Conclusions

In summary, we designed and developed a multifunctional naphthimide-based fluorescent probe (**MMN**). Because of the H^+^-induced intramolecular charge transfer (ICT) effect, **MMN** displayed significant pH sensitivity with apparent p*K_a_* values of 4.62 ± 0.02 and 5.43 ± 0.02 as well as good linear apparent pH responses ranging from 4.2 to 5.0 and from 5.0 to 6.0, which were suitable for selectively detecting acidic lysosomes. **MMN** shows potential for the rapid detection of pH in test papers. Furthermore, **MMN** could quickly and easily detect isoxaflutole with an extremely low detection limit. Density function theory calculations revealed MMN’s potential underlying the sensing mechanism.

## Data Availability

Not applicable.

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
