# Peer review of "A Multifunctional and Fast-Response Lysosome-Targetable Fluorescent Probe for Monitoring pH and Isoxaflutole"

_ijms, 2022, doi:10.3390/ijms23116256_

Round 1
Reviewer 1 Report
Similar ideas have been widely reported elsewhere, and hence the work lacks novelty. The chemosensor for monitoring pH and isoxaflutole has known limitations, which are not addressed in this paper. Most of results could be easily expected from what we have already known. The contribution of this study is limited. In addition, the mechanism study is poor.
Specific comments are as follows:
- Abstract. The problem statement and the implication of this study should be introduced.
- Introduction. The introduction of related previous studies on using chemosensor for monitoring pH and isoxaflutole should be added.
- Introduction Over 70% of papers cited were published 5 years ago, the recent advances and the research gap should be clearly introduced.
- This paper try to tell a full story, not a new story.
- The quality of figures should be improved.
- Related figures can be merged together.
- The data was presented without error to justify the repeatability of this study.
- The mechanism study is poor.
- Most of the results are presented without discussion.
Author Response
Answers for questions
For Referee 1
Similar ideas have been widely reported elsewhere, and hence the work lacks novelty. The chemosensor for monitoring pH and isoxaflutole has known limitations, which are not addressed in this paper. Most of results could be easily expected from what we have already known. The contribution of this study is limited. In addition, the mechanism study is poor.
- Abstract. The problem statement and the implication of this study should be introduced.
Author reply: Thank you for your helpful suggestions. We have tried our best to revise the abstract part in the manuscript.
Line 9: Abstract: A new chemosensor, namely N-(2-morpholinoethyl)acetamide-4-morpholine-1,8-naphthimide (MMN), was designed and synthesized through an amidation reaction. MMN was fabricated as a multifunctional fluorescent probe for monitoring pH and isoxaflutole. MMN exhibited excellent stability in MeCN/H2O (V/V, 9/1) with an obvious “off-on” fluorescence response toward pH changes due to intramolecular charge transfer (ICT), where the linear response ranges of MMN in the weak acidic system were from 4.2 to 5.0 and from 5.0 to 6.0 with apparent pKa= 4.62 ± 0.02 and 5.43 ± 0.02. Based on morpholine as the lysosome targetable unit, MMN could selectively locate lysosoms in live cells. MMN was also successfully applied to the detection of H+ in test paper. Finally, MMN could specifically recognize isoxaflutole with a detection limit of 0.88 μM and the possible sensing mechanism was identified by density function theory calculations. These results indicated that MMN could be a superior potential chemosensor for detecting pH and isoxaflutole selectively and sensitively and also could be used in real samples detection.
- Introduction. The introduction of related previous studies on using chemosensor for monitoring pH and isoxaflutole should be added.
Author reply: Thank you for your helpful suggestions. We have tried our best to revise the introduction part in the manuscript and added relative contents.
Line 24: The pH value is closely related to human life, which plays vital roles in some physiological and chemical and procedures, including enzyme activity, cell, environmental monitoring, human tissue fluid and so on [1].
Line 33: In the past few years, various techniques have been exploited for pH detection, such as colorimetric method [7], electrochemistry [8], ion sensitive field-effect transistors [9] and fluorescence techniques [10]. However, there are still shortcomings, such as photobleaching, high background emission, alkalization, complicated sample preparation, sophisticated instruments, time-consuming and high cost. Among them, fluorescent techniques have the advantages of convenience, efficiency, sensitivity and selectivity, which could be potential for monitoring pH.
Line 40: Pesticides refer to chemical agents used in agriculture to control pests and regulate plant growth. Pesticides are widely used in agriculture, forestry and animal husbandry, environmental and family hygiene, pest control and epidemic prevention, and industrial products to prevent mildew and moths [11].
Line 48: The current reports on the detection methods for isoxaflutole residues mainly involve liquid chromatography, liquid chromatography-tandem and mass spectrometry [18,19]. Even though, chromatography method has the merits of high sensitivity, superior separation ability and excellent selectivity, the disadvantages are long pretreatment time, interferences in the detection process, high costs and the combination with highly selective detectors for detecting pesticides. Herein, exploiting a more accurate, fast, efficient and convenient method for the detection of pesticides is needed. Chemosensors such as fluorescent probes may be potential materials for the detection of isoxaflutole due to the characteristics of low cost, easy operation, fast signal response and signal visualization.
- Introduction Over 70% of papers cited were published 5 years ago, the recent advances and the research gap should be clearly introduced.
Author reply: Thank you for your helpful suggestions. We have tried our best to revise the introduction part in the manuscript and added relative contents. The relative contents were listed in question 2.
References:
- J.B. Chao, H.J. Wang, Y.B. Zhang, Z.Q. Li, Y.H. Liu, F.J. Huo, C.X. Yin, Y.W. Shi, J.J. Wang, A single pH fluorescent probe for bio-sensing and imaging of extreme acidity and extreme alkalinity, Anal. Chim. Acta 2017, 975, 52–60.
- G.S. Luka, E. Nowak, J. Kawchuk, M. Hoorfar, H. Najjaran, Portable device for the detection of colorimetric assays, Roy. Soc. Open Sci. 2017, 4, 171025.
- L. Manjakkal, D. Szwagierczak, R. Dahiya, Metal oxides based electrochemical pH sensors: Current progress and future perspectives, Prog. Mater. Sci. 2020, 109, 100635.
- Y. Miao, J. Chen, K. Fang, New technology for the detection of pH, J. Biochem. Bioph. Meth. 2005, 63, 1–9.
- H. Zhou, R. Feng, Q. Liang, X. Su, L. Deng, L. Yang, L.J. Ma, A sensitive pH fluorescent probe based on triethylenetetramine bearing double dansyl groups in aqueous solutions and its application in cells, Spectrochim. Acta. A. 2021, 261, 120031.
- T. Kang, S. Gao, L.X. Zhao, Y. Zhai, F. Ye, Y. Fu, Design, synthesis and SAR of novel 1,3-disubstituted imidazolidine or hexahydropyrimidine derivatives as herbicide safeners, J. Agric. Food Chem. 2021, 69, 45–54.
- This paper try to tell a full story, not a new story.
Author reply: Thank you for your helpful suggestions.
The pH value is closely related to human life, which plays vital roles in some physiological and chemical and procedures, including enzyme activity, cell, environmental monitoring, human tissue fluid and so on. Meanwhile, the long-term wide applications of pesticides not only bring the potential health risks to humans, but also cause environmental pollution. On the basis of this point, the detection of pH and pesticides is urgent and necessary. Fluorescent probes may be potential materials for the detection of pH and isoxaflutole due to the characteristics of low cost, easy operation, fast signal response and signal visualization. Naphthalimide as a representative fluorophore, shows a high fluorescence quantum yield and high thermal stability. Morpholine has been recognized for its special lysosomal targeting function. In this work, MMN, as a novel fluorescent probe, was synthesized by introducing the morpholine unit to naphthalimide, which could specifically detect pH and isoxaflutole.
- The quality of figures should be improved.
Author reply: Thank you for your helpful suggestions. We have tried our best to improve the quality of figures and relative figures have been replaced in the manuscript.
- Related figures can be merged together.
Author reply: Thank you for your helpful suggestions. Related figures have been merged together in the manuscript and one figure was removed to supporting information.
Figure 2. (a) The linear responses of fluorescence intensity. (b) Reversibility of the fluorescence intensity of MMN between apparent pH 4.2 and 6.0. Inset: The color changes of MMN solution under UV light of 365 nm. (c) Changes in the fluorescence intensities of MMN with times among nonacidic and acid conditions.
Figure 3. (a) The 1H NMR spectra of probe MMN, MMN + H+ and MMN + OH- in D2O/DMSO-d6 (v/v, 1:1). (b) Proposed sensing mechanism of MMN toward H+.
Figure S5. The Job's plot of MMN with isoxaflutole determined by fluorescence spectra in CH3CN solution.
- The data was presented without error to justify the repeatability of this study.
Author reply: Thank you for your helpful suggestions. We have carefully checked all the data in the manuscript and ensured the repeatability of this study.
- The mechanism study is poor.
Author reply: Thank you for your helpful suggestions. We have tried our best to revise the manuscript carefully. Relative contents have been added to promote the mechanism study. The mechanism for detecting pH was investigated by 1H NMR of MMN (D2O/DMSO-d6 (V/V, 1:1)) were measured by the addition of TFA (H+) or NaOH (OH-) to solutions, respectively. The mechanism of MMN for detecting isoxaflutole was investigated by density functional theory (DFT) calculations due to the electron density or energy level between isoxaflutole and MMN.
Line 132: Therefore, when H+ was added, the change of 1H NMR spectrum contributed to the decrease in electron density around N-H of amide and the active hydrogen protons NH produced a charge transfer process, where N atom was positively charged and C=N was formed due to tautomerism.
Line 227: The transition of electrons from the LUMO orbital of MMN in the excited state to the LUMO of isoxaflutole prevented the electrons from returning to the HOMO orbital of MMN, leading to the phenomenon of fluorescence quenching for recognizing isoxaflutole. The electron-withdrawing group in isoxaflutole may play vital roles in quenching effects due to the photoinduced electron transfer from MMN to electron-deficient isoxaflutole. Therefore, DFT verified the possible mechanism of MMN for detecting isoxaflutole.
- Most of the results are presented without discussion.
Author reply: Thank you for your helpful suggestions. We have tried our best to revise the manuscript and added relative contents.
Line 74: MMN was supposed to exhibit an obvious “off-on” fluorescence response toward pH changes and could also recognize isoxaflutole.
Line 89: The results indicated that MMN displayed a superior “off-on” switching behavior from apparent pH 2 to 10 and an obvious change in the electron density of naphthalene rings, which was consistent with the fluorescent emission spectra analyses.
Line 109: These results indicated that MMN exhibited a high sensitivity to weak acidic pH with apparent pKa values of 4.62 ± 0.02 and 5.43 ± 0.02.
Line 115: Thus, it proved that MMN could be used as a reversible pH monitor due to its excellent reversibility.
Line 118: The results indicated that MMN could be reliably applied for real-time monitoring of pH in practical applications.
Line 203: The results implied that the fluorescence signal of free MMN remained stable, while MMN showed immediate and distinct fluorescence quenching, and the fluorescence intensity reached the minimum within 30 s and remained stable in the following 90 min, indicating the high reactivity of MMN with isoxaflutole.
Line 227: The transition of electrons from the LUMO orbital of MMN in the excited state to the LUMO of isoxaflutole prevented the electrons from returning to the HOMO orbital of MMN, leading to the phenomenon of fluorescence quenching for recognizing isoxaflutole. The electron-withdrawing group in isoxaflutole may play vital roles in quenching effects due to the photoinduced electron transfer from MMN to electron-deficient isoxaflutole. Therefore, DFT verified the possible mechanism of MMN for detecting isoxaflutole.

Reviewer 2 Report
The paper discussed A Multifunctional and Fast-Response Lysosome-Targetable Fluorescent Probe for Monitoring pH and Isoxaflutole. It was such an interesting subject to study. However, there are few things that i would appreciate authors to additionally describe in few sections in the manuscript. The manuscript seems to be suitable for publishing if revised according to the comments presented below:
“
- Novelty of this work should be written clearly.
- Introduction is short.
- Why N-(2-morpholinoethyl)acetamide-4-morpholine-1,8-naph- 9 thimide.?
- Could you clearly explain why cannot detect to concentration less than 0.88 µM?
- I think it is better if authors can calculate the efficiency of this type of pH at different rang of temperature.
- There are lot of typographical and grammatical mistakes. Even English language is not smooth and needs a thorough revision.
Author Response
Answers for questions
For Referee 2
The paper discussed A Multifunctional and Fast-Response Lysosome-Targetable Fluorescent Probe for Monitoring pH and Isoxaflutole. It was such an interesting subject to study. However, there are few things that i would appreciate authors to additionally describe in few sections in the manuscript. The manuscript seems to be suitable for publishing if revised according to the comments presented below:
- Novelty of this work should be written clearly.
Author reply: Thank you for your helpful suggestions. We have tried our best to revise the abstract part in the manuscript for the novelty of this work.
Line 9: A new chemosensor, namely N-(2-morpholinoethyl)acetamide-4-morpholine-1,8-naphthimide (MMN), was designed and synthesized through an amidation reaction. MMN was fabricated as a multifunctional fluorescent probe for monitoring pH and isoxaflutole. MMN exhibited excellent stability in MeCN/H2O (V/V, 9/1) with an obvious “off-on” fluorescence response toward pH changes due to intramolecular charge transfer (ICT), where the linear response ranges of MMN in the weak acidic system were from 4.2 to 5.0 and from 5.0 to 6.0 with apparent pKa= 4.62 ± 0.02 and 5.43 ± 0.02. Based on morpholine as the lysosome targetable unit, MMN could selectively locate lysosoms in live cells. MMN was also successfully applied to the detection of H+ in test paper. Finally, MMN could specifically recognize isoxaflutole with a detection limit of 0.88 μM and the possible sensing mechanism was identified by density function theory calculations. These results indicated that MMN could be a superior potential chemosensor for detecting pH and isoxaflutole selectively and sensitively and also could be used in real samples detection.
- Introduction is short.
Author reply: Thank you for your helpful suggestions. We have tried our best to revise the introduction part in the manuscript and added relative contents.
Line 24: The pH value is closely related to human life, which plays vital roles in some physiological and chemical and procedures, including enzyme activity, cell, environmental monitoring, human tissue fluid and so on [1].
Line 33: In the past few years, various techniques have been exploited for pH detection, such as colorimetric method [7], electrochemistry [8], ion sensitive field-effect transistors [9] and fluorescence techniques [10]. However, there are still shortcomings, such as photobleaching, high background emission, alkalization, complicated sample preparation, sophisticated instruments, time-consuming and high cost. Among them, fluorescent techniques have the advantages of convenience, efficiency, sensitivity and selectivity, which could be potential for monitoring pH.
Line 40: Pesticides refer to chemical agents used in agriculture to control pests and regulate plant growth. Pesticides are widely used in agriculture, forestry and animal husbandry, environmental and family hygiene, pest control and epidemic prevention, and industrial products to prevent mildew and moths [11].
Line 48: The current reports on the detection methods for isoxaflutole residues mainly involve liquid chromatography, liquid chromatography-tandem and mass spectrometry [18,19]. Even though, chromatography method has the merits of high sensitivity, superior separation ability and excellent selectivity, the disadvantages are long pretreatment time, interferences in the detection process, high costs and the combination with highly selective detectors for detecting pesticides. Herein, exploiting a more accurate, fast, efficient and convenient method for the detection of pesticides is needed. Chemosensors such as fluorescent probes may be potential materials for the detection of isoxaflutole due to the characteristics of low cost, easy operation, fast signal response and signal visualization.
- Why N-(2-morpholinoethyl)acetamide-4-morpholine-1,8-naph- 9 thimide.?
Author reply: Thank you for your helpful suggestions. The pH value is closely related to human life, which plays vital roles in some physiological and chemical and procedures, including enzyme activity, cell, environmental monitoring, human tissue fluid and so on. Meanwhile, the long-term wide applications of pesticides not only bring the potential health risks to humans, but also cause environmental pollution. On the basis of this point, the detection of pH and pesticides is urgent and necessary. Fluorescent probes may be potential materials for the detection of pH and isoxaflutole due to the characteristics of low cost, easy operation, fast signal response and signal visualization. Naphthalimide as a representative fluorophore, shows a high fluorescence quantum yield and high thermal stability. Morpholine has been recognized for its special lysosomal targeting function. In this work, MMN, as a novel fluorescent probe, was synthesized by introducing the morpholine unit to naphthalimide, which could specifically detect pH and isoxaflutole.
- Could you clearly explain why cannot detect to concentration less than 0.88 µM?
Author reply: Thank you for your helpful suggestions. Relative contents have been added in the manuscript.
Line 191: Moreover, based on the equation LOD = 3σ/k, the LOD of MMN for isoxaflutole was determined to be 0.88 µM, where σ is the standard deviation of the response at the lowest concentrations and k represents the slope of the calibration.
- I think it is better if authors can calculate the efficiency of this type of pH at different rang of temperature.
Author reply: Thank you for your helpful suggestions. In this work, MMN was used to detect pH in room temperature. Furthermore, we carried out the detection of MMN for pH in different temperature, such as 50 °C and 75 °C. Firstly, the stock solution of MMN was prepared in CH3CN/H2O (V/V, 9/1). Then, 10 ml of 1.00 × 10-5 M MMN solutions were placed in a water bath to maintain 50 °C and 75 °C. Lastly, 1.00 × 10-1 M HCl or NaOH was added to MMN solutions, and the pH value was determined with a pH meter. The fluorescence spectra of MMN exhibited a striking dependence on pH, and the maximum fluorescence intensities at 529 nm with different pH at 50 °C and 75 °C were nearly the same as MMN in room temperature. In summary, the effect of temperature on the probe MMN is small and can be ignored.
- There are lot of typographical and grammatical mistakes. Even English language is not smooth and needs a thorough revision.
Author reply: Thank you for your helpful suggestions. We have tried our best to revise our manuscript carefully and relative language retouching expert has been invited for eliminating typographical and grammatical errors in the full text. Relative certification for language retouch has been listed below.

Reviewer 3 Report
It is always nice to see the development of new fluorescent sensors, as well as the study of their properties. The work presented by Fu et al. is, in general terms, well presented and well written. The study has been logically designed and the methods properly described. However, I think it does not qualify for being published in IJMS due to its lack of novelty. The morpholine moiety has been extensively used as a lysosome-targeting unit. On the other hand, I do not see why the two morpholine subunits are essential to recognise isoxaflutole. Despite this, herein I provide some tips for the authors if they decide to present this work in a different journal:
1.- It would be nice to get a single crystal of MMN and discuss its structure.
2.- Figure S1: what is the pH or apparent pH of the ACN:water mixtures? This information is critical as, depending on pH, the compound might or might not be soluble (or not soluble enough) in these mixtures to record its spectra.
3.- Figure S2: what is the meaning of FEPTD? What is the solvent employed in these experiments? What about temperature? Please include all this information in those figures which require it.
4.- Lines 88-89: the authors should provide the errors of pKa values. A speciation diagram would also be desirable. Could the authors indicate to which morpholine corresponds each of these pKa values? I understand that they have been calculated in a 9:1 ACN:water mixture (please confirm this both in the text and figures), and in that case it would not be correct to say "pKa", but "apparent pKa". The same applies to pH values.
5.- Lines 115-116: "The 1H NMR spectrum was ascribed to the electron density around NH". I think this sentence should be re-written.
6.- Scheme 2: The mechanism is not convincing. Why do the authors assume that protonation occurs at the amide's N-H fragment? I think it is much more likely that it takes place at the carbonyl's oxygen atom. Then, a canonical form in which the nitrogen atom is positively charged would be possible (C-N double bond). Anyway, in the first case the nitrogen atom would be doubly protonated. And, at pH 4.2, one of the morpholines would be protonated, according to the apparent pKa values determined by the authors.
7.- Lines 136-137: the colours stated by the authors are not properly seen in the picture.
8.- Line 157: please assign this band and indicate the conditions in which these spectra were recorded (concentrations, solvent, temperature...).
9.- Line 165: after adding...
10.- Line 169: I would say: "a fluorescence titration of MMN was carried out in the presence of various concentrations of isoxaflutole". Please provide the structure of isoxaflutole, at least in SI.
11.- Line 175: the authors should indicate the meaning of sigma and k.
12.- Line 179: again, the authors should point out the meaning of each parameter. What is the error of K? Could this value be compared with others?
13.- Figure 9a: it is almost impossible to see the equations in the inset. I would also recommend the authors to increase the quality of the images displayed in the article.
14.- Line 202: do the authors mean HOMO?
15.- Line 264: DIPEA is not a coupling agent, but a base.
16.- Line 272: dd have only two coupling constants. Why do the authors provide three?
17.- Line 274: according to Figure S12, the signal at 2.36 ppm is not a doublet, but a multiplet. Please check all 1H and 13C NMR data.
18.- Lines 313-314: these apparent pKa values differ from those displayed in the abstract and in the text. What are the correct values?
General remarks: the authors should integrate all signals in 1H NMR spectra and provide HR mass spectra for all products, not only MMN. They should also provide DEPT-135 spectra. Could the authors revise the spectrum shown in Figure S7? I would expect more carbon signals for compound 1.
Author Response
Answers for questions
For Referee 3
It is always nice to see the development of new fluorescent sensors, as well as the study of their properties. The work presented by Fu et al. is, in general terms, well presented and well written. The study has been logically designed and the methods properly described. However, I think it does not qualify for being published in IJMS due to its lack of novelty. The morpholine moiety has been extensively used as a lysosome-targeting unit. On the other hand, I do not see why the two morpholine subunits are essential to recognise isoxaflutole. Despite this, herein I provide some tips for the authors if they decide to present this work in a different journal:
- It would be nice to get a single crystal of MMN and discuss its structure.
Author reply: Thank you for your helpful suggestions. We have tried our best to get the single crystal of MMN, while we failed. IR, 1H NMR, 13C NMR and mass Spectrum (M+H+) of MMN were all investigated and listed in supporting information.
- Figure S1: what is the pH or apparent pH of the ACN:water mixtures? This information is critical as, depending on pH, the compound might or might not be soluble (or not soluble enough) in these mixtures to record its spectra.
Author reply: Thank you for your helpful suggestions. We have added relative contents in the manuscript. MMN was soluble in CH3CN/H2O (V/V, 9/1) or in different pH solutions.
Line 81: Based on the solvent fluorescence response and low toxicity, the spectral properties of MMN were investigated in CH3CN/H2O (V/V, 9/1), where the apparent pH was 6.0.
- Figure S2: what is the meaning of FEPTD? What is the solvent employed in these experiments? What about temperature? Please include all this information in those figures which require it.
Author reply: Thank you for your helpful suggestions and relative contents have been added and revised in the supporting information. I am so sorry to make such a mistake, and “FEPTD” should be replaced by “MMN”.
Figure S1. Solvent effect (a) and fluorescence intensity of different proportions of acetonitrile and water (b) of MMN at room temperature.
Figure S2. (a) The fluorescence spectra of MMN with or without various metal ions in CH3CN/H2O (V/V, 9/1) at pH 4.2 at room temperature.
Figure S3. UV-Vis spectra change of MMN (10 µM) in CH3CN solution with or without isoxaflutole (50 µM) at room temperature.
Figure S4. Fluorescence intensity of MMN in the absence and presence of isoxaflutole in CH3CN solution.
Figure S5. The Job's plot of MMN with isoxaflutole determined by fluorescence spectra in CH3CN solution.
- Lines 88-89: the authors should provide the errors of pKa values. A speciation diagram would also be desirable. Could the authors indicate to which morpholine corresponds each of these pKa values? I understand that they have been calculated in a 9:1 ACN:water mixture (please confirm this both in the text and figures), and in that case it would not be correct to say "pKa", but "apparent pKa". The same applies to pH values.
Author reply: Thank you for your helpful suggestions. The errors of pKa values have been added in the manuscript, and corresponding “pKa” and “pH” were revised to “apparent pKa” and “apparent pH”. The calculation conditions of pKa were added in the manuscript.
Line 104: The fluorescence intensity of MMN with different apparent pH values was used to calculate the acidity constant apparent pKa values of MMN by Henderson-Hasselbalch equation in MeCN/H2O (V/V, 9/1).
- Lines 115-116: "The 1H NMR spectrum was ascribed to the electron density around NH". I think this sentence should be re-written.
Author reply: Thank you for your helpful suggestions. We have re-written this sentence.
Line 132: Therefore, when H+ was added, the change of 1H NMR spectrum contributed to the decreas in electron density around N-H of amide and the active hydrogen protons NH produced a charge transfer process, where N atom was positively charged and C=N was formed due to tautomerism.
- Scheme 2: The mechanism is not convincing. Why do the authors assume that protonation occurs at the amide's N-H fragment? I think it is much more likely that it takes place at the carbonyl's oxygen atom. Then, a canonical form in which the nitrogen atom is positively charged would be possible (C-N double bond). Anyway, in the first case the nitrogen atom would be doubly protonated. And, at pH 4.2, one of the morpholines would be protonated, according to the apparent pKa values determined by the authors.
Author reply: Thank you for your helpful suggestions. Relative contents and Figure 3 have been revised in the manuscript. While the possible mechanism was carried out in weak acidic system, the two morpholines were not protonated due to the 1H NMR spectra of MMN measured by the addition of TFA (H+) or NaOH (OH-) to solutions, respectively.
Line 132: Therefore, when H+ was added, the change of 1H NMR spectrum contributed to the decrease in electron density around N-H of amide and the active hydrogen protons NH produced a charge transfer process, where N atom was positively charged and C=N was formed due to tautomerism.
Figure 3. (a) The 1H NMR spectra of probe MMN, MMN + H+ and MMN + OH- in D2O/DMSO-d6 (v/v, 1:1). (b) Proposed sensing mechanism of MMN toward H+.
- Lines 136-137: the colours stated by the authors are not properly seen in the picture.
Author reply: Thank you for your helpful suggestions. We have tried our best to improve the quality of figures and relative figures have been replaced in the manuscript.
- Line 157: please assign this band and indicate the conditions in which these spectra were recorded (concentrations, solvent, temperature...).
Author reply: Thank you for your helpful suggestions. We have added relative contents in Figure S3.
Figure S3. UV-Vis spectra change of MMN (10 µM) in CH3CN solution with or without isoxaflutole (50 µM) at room temperature.
- Line 165: after adding...
Author reply: Thank you for your helpful suggestions. We have added relative contents in the manuscript.
Line 182: Figure 6. (a) Fluorescence spectra changes of the MMN solution (10 µM) after adding different pesticides (50 µM). Inset: color changes of MMN solution before and after adding pesticides under UV light of 365 nm. (b) Competition selectivity of MMN (10 µM) toward isoxaflutole (50 µM) in the presence of other competition pesticides (100 µM).
- Line 169: I would say: "a fluorescence titration of MMN was carried out in the presence of various concentrations of isoxaflutole". Please provide the structure of isoxaflutole, at least in SI.
Author reply: Thank you for your helpful suggestions. We have added relative contents in Figure 6.
Figure 6. (a) Fluorescence spectra changes of the MMN solution (10 µM) after adding different pesticides (50 µM). Inset: color changes of MMN solution before and after adding pesticides under UV light of 365 nm. (b) Competition selectivity of MMN (10 µM) toward isoxaflutole (50 µM) in the presence of other competition pesticides (100 µM).
- Line 175: the authors should indicate the meaning of sigma and k.
Author reply: Thank you for your helpful suggestions. We have added relative contents in the manuscript.
Line 191: Moreover, based on the equation LOD = 3σ/k, the LOD of MMN for isoxaflutole was determined to be 0.88 µM, where σ is the standard deviation of the response at the lowest concentrations and k represents the slope of the calibration.
- Line 179: again, the authors should point out the meaning of each parameter. What is the error of K? Could this value be compared with others?
Author reply: Thank you for your helpful suggestions and relative contents have been revised in the manuscript.
Line 195: The binding constant of MMN with isoxaflutole was calculated according to the fluorescence intensity data using the modified Stern-Volmer equation: [31], where I0 and I are the maximum luminescent intensities of MMN before and after adding isoxaflutole, K is the binding constant (M−1) and the unit measurement [A] represents molar concentration. The binding constant (K) was calculated as 3.7 × 105 M−1 (Figure 7b), which was compared with other previously reported fluorescent probes (Table S1).
Table S1. The comparison of binding constants with other reported fluorescent probes.
|
Type |
Probe |
Binding constant (K) (M-1) |
Detecting pesticides |
Ref. |
|
Off-on |
MMN |
3.7×105 |
Isoxaflutole |
This work |
|
On-off |
Rho B@1 |
9.08×104 |
nitenpyram |
[1] |
|
Off-on |
Tb@UiO-66 |
5.36×102 |
teflubenzuron |
[2] |
|
On-off |
RhB@Zr-MOF |
9.01×104 |
nitenpyram |
[3] |
References:
- L. Yang, Y.L. Liu, C.G. Liu, F. Ye, Y. Fu, Two luminescent dye@MOFs systems as dual-emitting platforms for efficient pesticides detection, J. Hazard. Mater. 2020, 381, 120966.
- L. Yang, Y.L. Liu, X.X. Ji, C.G. Liu, Y. Fu, F. Ye, A novel luminescent sensor based on Tb@UiO-66 for highly detecting Sm3+ and teflubenzuron, J. Taiwan Inst. Chem. Eng. 2021, 126, 173–181.
- L. Yang, Y.L. Liu, C.G. Liu, Y. Fu, F. Ye, A built-in self-calibrating luminescence sensor based on RhB@Zr-MOF for detection of cations, nitro explosives and pesticides, RSC Adv. 2020, 10, 19149–19156.
- Figure 9a: it is almost impossible to see the equations in the inset. I would also recommend the authors to increase the quality of the images displayed in the article.
Author reply: Thank you for your helpful suggestions. We have tried our best to improve the quality of figures and relative figures have been replaced in the manuscript.
- Line 202: do the authors mean HOMO?
Author reply: Thank you for your helpful suggestions. In the manuscript, “It was found that the LUMO energy levels of isoxaflutole and mesotrione were lower than those of other pesticides (glyphosate, oxyfluorfen, pyrazoxyfen, and cypermethrin), indicating that the electron affinities of isoxaflutole with the -CF3 electron-withdrawing group and mesotrione with the -NO2 electron-withdrawing group were relatively higher than those of other pesticides.” It means that the electron affinities of isoxaflutole and mesotrione were higher than the other pesticides, not HOMO.
- Line 264: DIPEA is not a coupling agent, but a base.
Author reply: Thank you for your helpful suggestions. We have revised relative contents to avoid ambiguity.
Line 280: Compound 2 (5 mmol) was reacted with 4-(2-aminoethyl)morpholine by the coupling reagents 1-ethyl-3-(3-dimethylaminopropyl)carbodiimide (EDC, 1.1 eq), the base N,N-diisopropylethylamine (DIEA, 2 eq), and hydroxybenzotrizole (HOBt, 1.2 eq) in dry DMF.
- Line 272: dd have only two coupling constants. Why do the authors provide three?
Author reply: Thank you for your helpful suggestions. We have dealt with the data of 1H NMR of MMN again.
Line 289: 1H NMR (DMSO-d6, TMS, 400 MHz, ppm) δ 8.54 (d, J = 8.5 Hz, 1H), 8.50 (d, J = 7.3 Hz, 1H), 8.43 (d, J = 8.1 Hz, 1H), 8.12 (s, 1H), 7.85 (s, 1H), 7.39 (m, 1H), 4.63 (s, 2H), 3.93 (t, J = 4.6 Hz, 4H), 3.56 (t, J = 4.7 Hz, 4H), 3.29 – 3.14 (m, 6H), 2.35 (m, 6H).
- Line 274: according to Figure S12, the signal at 2.36 ppm is not a doublet, but a multiplet. Please check all 1H and 13C NMR data.
Author reply: Thank you for your helpful suggestions. We have checked all 1H and 13C NMR data and dealt with the data of 1H NMR of MMN again.
Line 289: 1H NMR (DMSO-d6, TMS, 400 MHz, ppm) δ 8.54 (d, J = 8.5 Hz, 1H), 8.50 (d, J = 7.3 Hz, 1H), 8.43 (d, J = 8.1 Hz, 1H), 8.12 (s, 1H), 7.85 (s, 1H), 7.39 (m, 1H), 4.63 (s, 2H), 3.93 (t, J = 4.6 Hz, 4H), 3.56 (t, J = 4.7 Hz, 4H), 3.29 – 3.14 (m, 6H), 2.35 (m, 6H).
Figure S13. 1H NMR spectrum of compound MMN.
- Lines 313-314: these apparent pKa values differ from those displayed in the abstract and in the text. What are the correct values?
Author reply: Thank you for your helpful suggestions and relative contents have been revised in the manuscript. I am so sorry to make such a mistake.
Line 330: MMN displayed a significant pH-sensitive feature with pKa values of 4.62 ± 0.02 and 5.43 ± 0.02.
- General remarks: the authors should integrate all signals in 1H NMR spectra and provide HR mass spectra for all products, not only MMN. They should also provide DEPT-135 spectra. Could the authors revise the spectrum shown in Figure S7? I would expect more carbon signals for compound 1.
Author reply: Thank you for your helpful suggestions. The HR and DEPT-135 were failed to provide. While, we have integrated all signal in 1H NMR, and 13C NMR spectrum of compound 1, 2 and MMN were also revised to display more carbon signals. I think the IR, 1H NMR, 13C NMR and HR spectra could identify the successful synthesis of compound 1, 2 and MMN.
Line 261: 1H NMR (CDCl3, TMS, 400 MHz, ppm) δ 8.61 (m, 1H), 8.55 (d, J = 8.1 Hz, 1H), 8.48 (m, 1H), 7.76 (m, 1H), 7.28 (s, 1H), 4.07 – 4.00 (m, 4H), 3.35 – 3.29 (m, 4H).
Line 274: 1H NMR (DMSO-d6, TMS, 400 MHz, ppm) δ 13.04 (s, 1H), 8.57 – 8.42 (m, 3H), 7.85 (m, 1H), 7.39 (d, J = 8.1 Hz, 1H), 4.72 (s, 2H), 3.97 – 3.84 (m, 4H), 3.29 – 3.19 (m, 4H).
Line 289: 1H NMR (DMSO-d6, TMS, 400 MHz, ppm) δ 8.54 (d, J = 8.5 Hz, 1H), 8.50 (d, J = 7.3 Hz, 1H), 8.43 (d, J = 8.1 Hz, 1H), 8.12 (s, 1H), 7.85 (s, 1H), 7.39 (m, 1H), 4.63 (s, 2H), 3.93 (t, J = 4.6 Hz, 4H), 3.56 (t, J = 4.7 Hz, 4H), 3.29 – 3.14 (m, 6H), 2.35 (m, 6H).
Figure S7. 1H NMR spectrum of compound 1.
Figure S8. 13C NMR spectrum of compound 1.
Figure S10. 1H NMR spectrum of compound 2.
Figure S11. 13C NMR spectrum of compound 2.
Figure S13. 1H NMR spectrum of compound MMN.
Figure S14. 13C NMR spectrum of compound MMN.

Round 2
Reviewer 1 Report
It can be accepted.
Reviewer 3 Report
As I said in my previous comments, I think this work is not suitable for publication in IJMS due to its lack of novelty. I encourage the authors to try a different journal.